REGISTERED REPORT PROTOCOL

# Effect of MBSR, DBT and CBT on the hypertension patients with depression/anxiety: Protocol of a systematic review and Bayesian network meta-analysis

Yan Zhang[1☯], Hailiang Zhang[2,3☯], Yong Zhang[4], Zijiao Yang[5], Lingling Wang[6], Weimin Pan[3], Runjing Dai[3], Qianqian Ju[3], Dong Ren[7]*, Shisan Bao[3]*, Jingchun Fan[3]*

1 School of Basic Medicine, Gansu University of Chinese Medicine, Lanzhou, Gansu, China, 2 Department of Mental Health, Gansu Provincial Centre for Disease Control and Prevention, Lanzhou, Gansu, China, 3 School of Public Health, Center for Laboratory and Simulation Training, Center for Evidence-based Medicine, Gansu University of Chinese Medicine, Lanzhou, Gans, China, 4 Health Center of Dachaigou Town, Wuwei, Gansu, China, 5 West China School of Medicine, Sichuan University, Chengdu, Sichuan, China, 6 Community Health Service Center of Caochang Street, Lanzhou, Gansu, China, 7 Psychosomatic and Sleep Medicine, Gansu Gem Flower Hospital, Lanzhou, Gansu, China

☯ These authors contributed equally to this work.
* fan_jc@126.com (JF); profbao@hotmail.com (SB); lzshld@163.com (DR)

## Abstract

### Introduction

Hypertension, one of the most common chronic diseases worldwide, usually requires lifetime managing blood pressure (BP) with medications. Due to quite large number of hypertension patients co-exist with depression and/or anxiety, and non-cooperated with medical instruction, consequently management of BP is impaired with serious complications, resulting in compromised quality of life. Consequently quality of life of such patients is impaired with serious complications. Therefore, management of depression and/or anxiety is equally important as the treatment of hypertension. Depression and/or anxiety are independent risk factors of hypertension, which is supported by the finding that there is close correlation between hypertension are depression/or anxiety. Psychotherapy (non-drug treatment) maybe useful for hypertensive patients with depression and/or anxiety to improve their negative emotions. We aim to quantify the effective of psychological therapies in the management of hypertension patients with depression or anxiety, by comparing and ranking a network meta-analysis (NMA).

### Materials and methods

Literature search for randomized controlled trials (RCTs) will be performed in five electronic databases from inception to December 2021, including PubMed, the Cochrane library, Embase, Web of Science, and China Biology Medicine disc (CBM). The search terms mainly include "hypertension", "mindfulness-based stress reduction" (MBSR), "cognitive behavioral therapy" (CBT) and "dialectical behavior therapy" (DBT). Cochrane Collaboration quality assessment tool will be used for the risk of bias assessment. A Bayesian network

**Data Availability Statement:** All relevant data from this study will be made available upon study completion.

**Funding:** The authors are grateful to Dr Jinhui Tian, The Center for Evidence-Based Medicine, Lanzhou University for helpful suggestions on search strategy and data analysis. This work is supported by The 2020 Science and Technology Project of Chengguan District, Lanzhou [grant number 2020-2-11-16], The Talent Introduction Program of Gansu University of Chinese Medicine, Gansu University of Chinese Medicine [grant number 2016YJRC-01] and The University Innovation Capacity Improvement Project in Gansu Province [grant number 2020B-153].

**Competing interests:** The authors have declared that no competing interests exist.

**Abbreviations:** MBSR, Mindfulness-based stress reduction; CBT, Cognitive behavior therapy; DBT, Dialectical behavior therapy; RCTs, Randomized controlled trials; BP, blood pressure; OR, Odds ratio; WMD, Weighted mean difference; 95%CI, 95% Credible intervals; PSRF, Potential scale reduction factors.

meta-analysis will be performed, using WinBUGS 1.4.3, and Stata 14 will be applied to draw the network diagram, while RevMan 5.3.5 will be used to produce funnel plot for assessing the risk of publication bias. Recommended rating, development and grade methodology will also be utilized to assess the quality of evidence.

## Results

Effect of MBSR, CBT and DBT will be evaluated by traditional meta-analysis directly and Bayesian network meta-analysis indirectly. Our study will provide the evidence on the efficacy and safety of psychological treatments in the hypertension patients with anxiety. There is no research ethical requirement because this is a systematic review of published literature. The results of this study will be published in a peer-reviewed journal.

## Trial registration

**Prospero registration number:** CRD42021248566.

## 1. Introduction

It is predicted that the number of hypertensive adults will reach 1.5 billion, which is ~ 30% of the world population by 2025, based on analysis of worldwide data of hypertension burden [1]. Hypertension is notoriously difficult to control, causing detrimental irreversible damage to the important system in the body, e.g. central nerves system and cardiovascular system. In addition to physical impact, psychological disturbance is another major concern among these uncontrolled hypertension patients [2]. Rapid economic development and fast globalization perhaps contribute to the increased stress to the general population *via* transforming our social organization [3]. The literature has accumulated evaluating the effect of psychological stress in cardiovascular field, including depression and anxiety [4, 5], which are mutually causal and affect each other, and further aggravate the psychological impairment. There are some reports, showing the efficacy of psychological intervention for the management of the patients with comorbidity of hypertension and depression/anxiety. We will explore/validate such findings, using meta-analysis in current study.

It has been reported that substantially increased volume of psychological intervention for the management of hypertension patients with depression and/or anxiety over the last two decades [6]. Depression and anxiety are significant contributors to hypertension at the global level with huge economic burden. The World Mental Health Survey from 17 countries demonstrates that on average about 1 in 20 people reported having an episode of depression in the previous experience [7], suggesting common problem nowadays we face. It has reported that the incidence of depression in hypertension patients reached 20%, which is supported by the finding that cardiovascular disease patients have a higher incidence of psychological disorders, accompanied with increased the risk of cerebrovascular accident in the general population with anxiety and depression disorder [8]. This is supported by the finding from Netherlands for a 5-year study involving the 455,238 women, confirming a close correlation between hypertension and depression/anxiety. Furthermore, depression or anxiety increases 3.5 or 2.0 fold of the risk of hypertension from non-hypertensive people [9]. The finding from Netherlands is consistent with other regions, showing that depression or anxiety increases the risk of hypertension by 1.42 [10] or 1.55 [11] times. Patients with depression and anxiety usually presented

with dizziness, headache and chest tightness [12], and often accompanied with negative emotions which impacted quality of life. Such phenomenon is attracted great attention from psychologists recognizing the needs from mental unstable patients [13].

Effective significant increases for psychological therapies have been seen in hypertension patients with depression and anxiety in the past two decades. Due to inferences from similar study are somewhat constrained methodological issues, including no control group, inclusion of measurement of treatment adherence, or measurement of follow-up outcomes, and inclusion of patients with concurrent use of psychotropic medications, and unequal duration of treatment. To date, few studies have directly addressed whether different psychotherapies or similar treatments have comparable efficacy for treatment of hypertension patients with depression and anxiety. Several clinical practice guidelines recommend that in hypertension patients, psychotherapy should be considered as the first-line intervention for the management of hypertension patients with depression and anxiety disorder, especially suitable when anti-depression or anti-anxiety drugs are not working or not available [14, 15]. The evidence-based for various psychotherapy to be more effective and safer in the treatment of depression or anxiety disorder in hypertension patients is not well established [16]. Most of the meta-analysis published are conducted a direct comparison of the various interventions, rather than an indirect comparison of the interventions. Bayesian network meta-analysis (NMA) has the advantage that all interventions that have been tested in randomized controlled trials (RCTs) can be simultaneously compared, without requiring direct within-study treatment versus treatment comparisons. It is reported that mindfulness-based stress reduction (MBSR) [17, 18] and cognitive behavior therapy (CBT) [19] have a good effect in treating chronic diseases patients, and dialectical behaviour therapy (DBT) [20] has a significant effect in management of depression and anxiety. In this protocol, we intend to examine if these interventions improve depression, anxiety and blood pressure from these cohorts. We will incorporate the relevant literature of the above three therapies, and use Bayesian network meta-analysis to explore the relative effect and/or safety between different psychotherapies with depression relieved as the primary outcome index, anxiety relieved and blood pressure improved as the secondary outcome index.

## 2. Methods/Design

### 2.1 Research objectives

The aim of the current protocol is to synthesize all psychotherapies evidence and to provide clinicians with a reliable and optimal treatment for depressive and/or anxiety disorder in hypertension patients.

### 2.2 Inclusion criteria

**2.2.1 Types of studies.** We will include only randomized controlled trials (RCTs) with MBSR, CBT, DBT.

**2.2.2 Types of participants.** 1) Hypertension patients with depression or anxiety; 2) Hypertension patients who volunteer for psychotherapy; 3) inpatient/outpatient.

**2.2.3 Types of interventions.** The eligible interventions include MBSR, CBT, and DBT, the length of each intervention is 8–10 weeks, and the follow-up time is 1–3 months.

**2.2.4 Types of comparators.** The control interventions will be included drug therapy and exercise therapy (Yoga, Tai Chi, etc.).

**2.2.5 Types of outcomes.** The primary outcome indexes of the protocol are depression and anxiety score, and the secondary outcome indexes mainly include blood pressure and other outcome indexes reported in the literature.

## 2.3 Exclusion criteria

The exclusion criteria include:

1. Hypertension patients who receive any combination of two or more therapies other than MBSR, CBT and DBT.

2. Any hypertension patients treated with any psychotherapy other than the above.

3. Case reports, reviews, abstracts, experimental studies, mechanism discussions, experience summary and other research types of literature.

4. Repeatedly checked or published literature.

5. Incomplete data or information that does not indicate the end result and cannot be included.

## 2.4 Information sources and search strategy

A literature search for RCTs will be performed in five electronic databases from inception to December 2022, including: PubMed, the Cochrane Library, EMBASE, Web of Science and China Biology Medicine disc. In order to find more relevant papers, we will conduct forward and backward citation screening through the citation and bibliography of systematic review. Multiple synonyms for each word will be incorporated into the search. Search strategy of PubMed is provided in Table 1.

## 2.5 Register

The protocol of the present study was registered in the international prospective register of systematic reviews (Prospero registration number: CRD42021248566).

## 2.6 Date management

The search results will be imported to the document management software (Endnote X9). Before the formal selection of literature, we will conduct two inspections after the literature is

**Table 1. Search strategy (PubMed).**

| #1 | Search "Hypertension"[Mesh] |
|---|---|
| #2 | Search (High Blood Pressure*[Title/Abstract]) OR (Hypertension[Title/Abstract]) |
| #3 | Search #1 OR #2 |
| #4 | Search "mindfulness-based stress reduction" [MeSH] |
| #5 | Search mindfulness-based stress reduction[Title/Abstract] |
| #6 | Search #4 OR #5 |
| #7 | Search "Cognitive Behavioral Therapy"[Mesh] |
| #8 | Search ((((Cognitive Behavioral Therap*[Title/Abstract]) OR (Cognitive Behavior Therap*[Title/Abstract])) OR (Cognitive Therap*[Title/Abstract])) OR (Cognitive Psychotherap*[Title/Abstract])) OR (Cognition Therap*[Title/Abstract]) |
| #9 | Search #7 OR #8 |
| #10 | Search "Dialectical Behavior Therapy"[Mesh] |
| #11 | Search Dialectical Behavior Therap*[Title/Abstract] |
| #12 | Search #10 OR #11 |
| #13 | Search #6 OR #9 OR #12 |
| #14 | Search #3 AND #13 |
| #15 | Search #14 Filters: Randomized Controlled Trial |

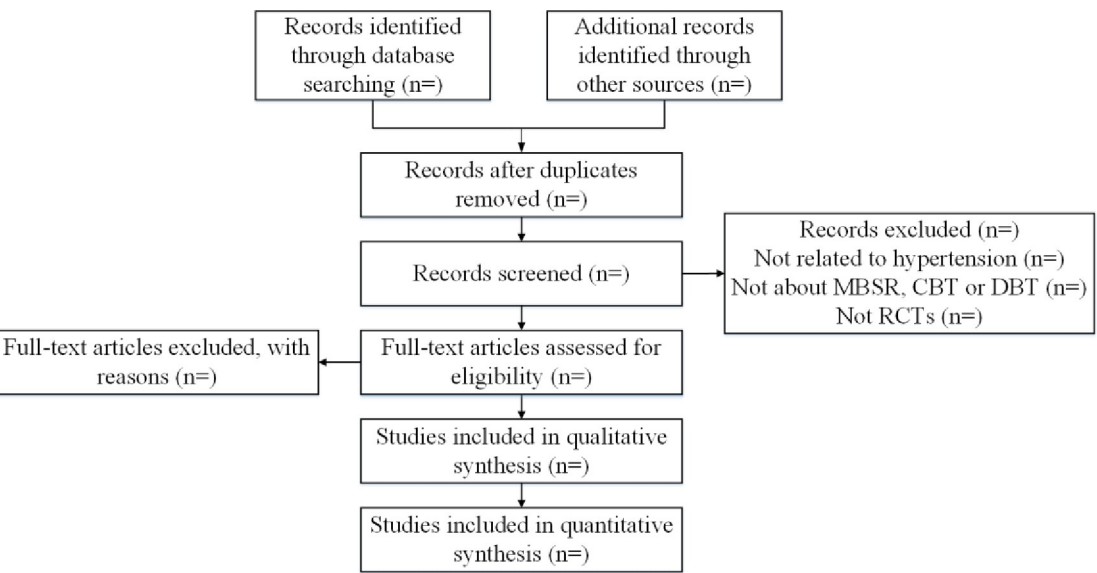

**Fig 1. Preferred reporting items for systematic review and Bayesian network meta-analysis.**

exported and the duplicate literature is excluded. The software will record our screening process.

## 2.7 Study screening

Two reviewers will screen the articles independently, based on the titles/abstracts and full texts. A final judgment will be made following the discussion with the third senior reviewer, in case of disagreement. If the study data is repeated, only the studies with large sample size and long follow-up time will be included. The flow chart of study selection is displayed in Fig 1.

## 2.8 Data extraction

Two independent reviewers will extract the data from each included trial, using standardized data extraction forms, including study characteristics (e.g., all listed authors, publication year, title, publication type, publication journal, country and sponsor), characteristics of the patients (e.g., diagnostic criteria, comorbidities, the age, setting, number of sample, sex, and severity of depression at baseline), intervention details (e.g., the type of intervention, the treatment duration, the length and number of sessions of psychotherapy, treatment delivery and treatment medium of psychotherapy) and outcome measures (primary outcomes and secondary outcomes). We will assess and report the reliability of the reviewers' data extraction on each coded variable. Any disagreements will be resolved by a third review reviewers, as described above. Whenever it is necessary, the authors of the studies will be contacted for further information.

## 2.9 Outcomes

We will utilize the data available from the selected literatures, but not only narrow our study into the PHQ-9 depression scale and the GAD-7 anxiety scale. The primary outcomes are the modified depression and/or anxiety scores, by comparing prior to and post intervention. We will further determine the blood pressure prior to and post intervention from these cohorts, as objective evidence for improving the patients' conditions. The treatment duration will be

defined as from 8 to 10 week. We will exclude trials with treatment duration of less than 8 weeks, because the onset of benefit for most psychotherapies often takes at least 8 weeks. Furthermore, trials comparing the same type of psychological interventions, but at different numbers of therapeutic sessions, different delivery format (group, individual) [21], different treatment media (face-to-face, internet-based) [22, 23] and different treatment conditions will be considered as the same node in the network analysis. We anticipate that any patient(s) who meet all inclusion criteria, in principal, is equally likely to be randomized to any of the interventions in the synthesis comparator set.

The secondary outcome is efficacy (as dichotomous outcome), measured by the total number of patients who achieved the criteria of remission, defined as being below the threshold in blood pressure (e.g., less than 140/90 mmHg for blood pressure).

In addition, the outcome indicators also include any other outcomes mentioned in the original study.

## 2.10 Risk of bias assessment

The methodological quality of each included study will be assessed, using the Cochrane Collaboration Risk of Bias Tool (CCRBT), by two independent reviewers. The assessment tool includes the following criteria: random sequence generation, allocation concealment, blinding of participants and personnel, blinding of the results assessment, incomplete data of the results, selective reporting, and other sources of bias [24]. Disagreement will be solved by discussion with a third senior reviewer.

## 2.11 Statistical analysis

NMA combines direct and indirect evidence for all relative treatment effects and provides estimates with maximum power. First, we will perform pairwise meta-analyses of direct evidence using the random-effects model with Stata V.14.0. Second, we will also perform a random-effects NMA within a Bayesian framework, using Markov chain Monte Carlo in WinBUGS V.1.4.3. Where different measures are used to assess the same outcome, continuous outcomes data will be pooled with standardized mean difference (SMD) and dichotomous outcomes will be analyzed by calculating the Odds ratio (OR).

Missing dichotomous outcome data will be managed according to the intention to treat (ITT) principle, and all the dropouts after randomization will be considered to be non-responders. Missing continuous outcome data will be analyzed, using the completer data. When $P$ values, $t$ values, CI or SE are reported in articles, SD will be calculated from their values. If SD is/are missing, attempts will be made to obtain these data through contacting trial authors. When this fails, they will be borrowed from the other trials in the network or from other published reports.

Potential Scale Reduction Factors (PSRF) will be used to evaluate the model convergence. The closer PSRF will be to 1, the better the model convergence is. A two-tailed value of $P \leq 0.05$ will be considered to indicate statistical significance. We will use the test to estimate the presence of statistical heterogeneity with threshold as $P \leq 0.05$. The $I^2$ test will be used to estimate the degree of heterogeneity as truncating $I^2 \geq 50\%$ as considerable heterogeneity. The sensitivity analysis will be conducted if the heterogeneity is significant. The purpose of these studies will explore whether such factor(s) might have an impact on our results.

## 2.12 Confidence in cumulative estimate

We will use the recommended rating, development and rating methods to assess the quality of direct and indirect evidence. The quality of evidence will be graded as high, moderate or low.

The studies will be performed by two independent reviewers. If there are different opinions, the decision will be made after consultation with the third investigator.

### 2.13 Assessment of publication bias

Bergg's and Egger's tests will be used to help distinguish the asymmetry due to publication bias [25, 26].

## 3. Discussion

At present, MBSR, CBT and DBT are common psychotherapies in the management of the hypertension patients with depression and anxiety, and a sizeable proportion of hypertension patients are responded positively to those psychotherapies [27, 28]. However, the effect of MBSR, CBT, and DBT in the combination treatment of hypertension with depression and anxiety is still unclear. We have planned this systematic review to address this current knowledge gap.

The purpose of this study is to conduct a network meta-analysis on the MBSR, CBT and DBT, and to determine the relative effects and/or safety way to hypertension patients with depression and/or anxiety. Our study may provide hypertension patients with the reliable evidence on the efficacy and safety of psychotherapy, which might contribute to future clinical trials and study design. In the current study, we will only include RCTs. Of course, in order to ensure that the RCTs literature included is comprehensive, we will track and retrieve relevant references for published systematic evaluations and meta-analyses. In addition, although there are a large number of literatures on the effect of psychotherapies and other interventions, the number of literatures on RCTs may also be relatively small and the literature quality is not high, which may affect the results of this study.

## Supporting information

**S1 Checklist. PRISMA-P (preferred reporting items for systematic review and meta-analysis protocols) 2015 checklist: Recommended items to address in a systematic review protocol**\*.
(DOC)

## Acknowledgments

The authors are grateful to Dr Jinhui Tian, The Center for Evidence-Based Medicine, Lanzhou University for helpful suggestions on search strategy and data analysis.

## Author Contributions

**Data curation:** Lingling Wang, Dong Ren.

**Project administration:** Yong Zhang.

**Software:** Zijiao Yang, Weimin Pan, Runjing Dai, Qianqian Ju.

**Supervision:** Zijiao Yang.

**Validation:** Shisan Bao, Jingchun Fan.

**Writing – original draft:** Yan Zhang, Hailiang Zhang.

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
