## [Decision Letter · Decision Letter 0]

11 Aug 2022

PONE-D-21-36202Effect comparison of Psychotherapies on hypertension with depression and/or anxiety: Protocol of a systematic review and Bayesian network meta-analysisPLOS ONE

Dear Dr. Zhang,

Thank you for submitting your manuscript to PLOS ONE. After careful consideration, we feel that it has merit but does not fully meet PLOS ONE’s publication criteria as it currently stands. Therefore, we invite you to submit a revised version of the manuscript that addresses the points raised during the review process.

We look forward to receiving your revised manuscript.

Kind regards,

Stephan Doering, M.D.

Academic Editor

PLOS ONE

4. PLOS requires an ORCID iD for the corresponding author in Editorial Manager on papers submitted after December 6th, 2016. Please ensure that you have an ORCID iD and that it is validated in Editorial Manager. To do this, go to ‘Update my Information’ (in the upper left-hand corner of the main menu), and click on the Fetch/Validate link next to the ORCID field. This will take you to the ORCID site and allow you to create a new iD or authenticate a pre-existing iD in Editorial Manager. Please see the following video for instructions on linking an ORCID iD to your Editorial Manager account: https://www.youtube.com/watch?v=_xcclfuvtxQ.

Reviewers' comments:

Reviewer's Responses to Questions

**Comments to the Author**

1. Does the manuscript provide a valid rationale for the proposed study, with clearly identified and justified research questions?

Reviewer #1: Partly

Reviewer #2: No

2. Is the protocol technically sound and planned in a manner that will lead to a meaningful outcome and allow testing the stated hypotheses?

Reviewer #1: Partly

Reviewer #2: Partly

3. Is the methodology feasible and described in sufficient detail to allow the work to be replicable?

Reviewer #1: Yes

Reviewer #2: No

4. Have the authors described where all data underlying the findings will be made available when the study is complete?

Reviewer #1: No

Reviewer #2: Yes

5. Is the manuscript presented in an intelligible fashion and written in standard English?

Reviewer #1: No

Reviewer #2: No

6. Review Comments to the Author

You may also provide optional suggestions and comments to authors that they might find helpful in planning their study.

Reviewer #1: Thank you for the very interesting report protocol, which addresses an important issue.

However, from my point of view, publication at this stage is not yet reasonable due to following concerns:

1.) Improving scope and rigor of your argumentation is highly recommended:

- Starting with the research question in the title since you do not want to compare psychotherapies as such but rather specific methods (MBSR, DBT, CBT).

- Late in the text (2.8. outcomes), you make it clear that, in addition to examining improvements in psychological distress, you also want to analyze the potential reduction in blood pressure.

- The argumentative approach to the research question is vague. It is not clear whether you want to examine improvement in depression, anxiety, and blood pressure or of all three factors in your meta-analysis. It remains unclear and is not adequately supported with citations as to why you choose DBT, CBT, and MBSR.

- The argumentation why the research question is useful still seems a bit weak. You could base your argument on well-documented evidence and argue: High prevalence of hypertension in population; high prevalences of depression and anxiety in hypertension; associations of mental distress and important outcomes in hypertension (mortality, quality of life, disease progression, health behavior, utilization patterns, etc.). Next, the evidence for the efficacy of the therapeutic procedures studied, which you now want to examine with the meta-analysis.

2.) Please improve citation. Some of the cited references given are insufficient, misleading or incorrect. Examples:

- (1) Hu et al. (2015) is not the original source of the statement given. It'd rather be: Kearney PM, Whelton M, Reynolds K, Muntner P, Whelton PK, He J. Global burden of hypertension: analysis of worldwide data. Lancet. 2005 Jan 15-21;365(9455):217-23. doi: 10.1016/S0140-6736(05)17741-1. PMID: 15652604.

- (6) again the cited source seems to be not the original reference, it'd rather be World Health Organization

- The selection of therapeutic methods (DBT, CBT, MBSR) studied is not sufficiently supported with references. Note that source (17), to which your argument refers, that these treatments are "the most popular psychotherapies depression and/or anxiety in hypertension patients", seems to be a mere study protocol which doesn't mention DBT and MBSR at all.

- Please note that citation (27), which is supposed to support your statement: "At present, MBSR, CBT and DBT are common psychotherapies in the management of the hypertension patients with depression and anxiety..." refers to Combined Exercise Training, an apparently entirely different form of therapy.

- I could not find a reference to the DBT in your text

3.) You could also slightly sharpen the methodological description: for example, a more comprehensive and precise listing of the inclusion criteria would be useful (definition of the therapy methods studied e.g. length of interventions; inpatient/outpatient; outcomes collected in the included studies and their measurement, minimum sample size; follow-up-timing etc).

4.) Since the draft is a registered study protocol, I suspected that the searching and data analysis was still pending. In point 2.4 you write that the information search was finished in December 2021. Is this correct?

5.) Language: The submitted manuscript contains a number of grammatical errors and lacks comprehensibility in several places. For example, I do not understand the first sentence in the second paragraph of the introduction.

Reviewer #2: First, you should use proper English.

Using "will" is inappropriate in most parts.

It is appropriate to use the past tense for what you have analyzed.

You should also avoid writing in a way that makes it unclear whether it is your idea or a cited finding. In particular, there are a number of sentences that are described as general findings but do not have citations attached.

Major point

1. Introduction in abstract

The following is an exaggeration, because it is stated as if all hypertensive patients are suffering from depression and anxiety as well.

“However the outcomes of medications maybe compromised in some individuals without following medical instruction, partially due to these hypertension patients are co-exist with depression and/or anxiety.”

The following leads to the erroneous conclusion that measures to address the risk factor will improve hypertension.

“Because depression/or anxiety is an independent risk factor for hypertension, stress relieving would be an ideal management of hypertension. Psychotherapy for hypertension patients with depression and/or anxiety improves psychological symptoms and controls BP.”

2. Introduction

Please add a reference to the following sentence.

“In addition to physical disability, psychological disturbance is another major concerns among these uncontrolled hypertension patients. “

“Rapid development and fast globalization perhaps contribute to the increased stress level via transforming our social organization. ”

From the findings of the literature in [3], the following statements are clearly overstated

“It has been demonstrated that significantly increased psychological intervention for the hypertension patients with depression and/or anxiety over the last two decades [3].

”

3. Method

For example, please specify the Study protocol in the following form:

[The protocol of the present study was registered in the international prospective register of systematic reviews]

You have narrowed your criteria for depression and anxiety to two: the PHQ-9 depression scale and the GAD-7 anxiety scale. Do you have any clear explanatory evidence that this does not constitute selection bias?

4. Results.

　There is no results chapter.

For example, please describe in the following way.

How many cases were included in the inclusion criteria, how many were excluded, and how the results differed for each study.

5. Discussion

Discussion is meaningless because there is no result.

6. Figure

Please create a figure of the literature search flow chart showing the number of literature hits for each literature search method. If you create a figure, please also create network diagrams of comparisons.

Minor point

AGREE II(The Appraisal of Guidelines for Research & Evaluation II) and other tools, and consider whether a systematic review is sufficient in the first place or whether a meta-analysis is necessary.

7. PLOS authors have the option to publish the peer review history of their article (what does this mean?). If published, this will include your full peer review and any attached files.

Reviewer #1: No

Reviewer #2: No

---

## [Author Response · Author response to Decision Letter 0]

16 Dec 2022

Dr. Stephan Doering

Academic Editor, 

PLOS ONE

25 October 2022

Dear Dr Doering

We appreciate the constructive comments made by the reviewers and our responses are as follows: 

1. Does the manuscript provide a valid rationale for the proposed study, with clearly identified and justified research questions? 

Reviewer #1: Partly

Reviewer #2: No

Acknowledged.

2. Is the protocol technically sound and planned in a manner that will lead to a meaningful outcome and allow testing the stated hypotheses?

Reviewer #1: Partly

Reviewer #2: Partly

Acknowledged.

3. Is the methodology feasible and described in sufficient detail to allow the work to be replicable?

Reviewer #1: Yes

Reviewer #2: No

Acknowledged.

4. Have the authors described where all data underlying the findings will be made available when the study is complete?

Reviewer #1: No

Reviewer #2: Yes

Acknowledged.

5. Is the manuscript presented in an intelligible fashion and written in standard English?

Reviewer #1: No

Reviewer #2: No

Acknowledged.

6. Review Comments to the Author

Reviewer #1:

We appreciate the positive comment made by the reviewer “Thank you for the very interesting report protocol, which addresses an important issue.”

However, from my point of view, publication at this stage is not yet reasonable due to following concerns:

1.) Improving scope and rigor of your argumentation is highly recommended:

- Starting with the research question in the title since you do not want to compare psychotherapies as such but rather specific methods (MBSR, DBT, CBT).

Effect of MBSR, DBT and CBT on the hypertension patients with depression/anxiety: Protocol of a systematic review and Bayesian network meta-analysis

- Late in the text (2.8. outcomes), you make it clear that, in addition to examining improvements in psychological distress, you also want to analyze the potential reduction in blood pressure.

To clarify this point, we have modified our manuscript, it now reads: “The primary outcomes are the modified depression and/or anxiety scores, by comparing prior to and post intervention. We will further determine the blood pressure prior to and post intervention from these cohorts, as objective evidence for improving the patients’ conditions.” (Methods, line 176 paragraph 2 page10)

- The argumentative approach to the research question is vague. It is not clear whether you want to examine improvement in depression, anxiety, and blood pressure or of all three factors in your meta-analysis. It remains unclear and is not adequately supported with citations as to why you choose DBT, CBT, and MBSR.

We have added the following sentence for clarification it now reads: “In this protocol, we intend to examine if these interventions improve depression, anxiety and blood pressure from these cohorts”. In addition, we have updated our cited references accordingly, it now reads: “It is reported that mindfulness-based stress reduction (MBSR) [17, 18] and cognitive behavior therapy (CBT) [19] have a good effect in treating chronic diseases patients, and dialectical behaviour therapy (DBT) [20] has a significant effect in management of depression and anxiety.” (Introduction, line 104 paragraph 1 page 6)

- The argumentation why the research question is useful still seems a bit weak. You could base your argument on well-documented evidence and argue: 

a. High prevalence of hypertension in population; 

b. high prevalence of depression and anxiety in hypertension;

c. associations of mental distress and important outcomes in hypertension (mortality, quality of life, disease progression, health behavior, utilization patterns, etc.).

d. Next, the evidence for the efficacy of the therapeutic procedures studied, which you now want to examine with the meta-analysis.

Appreciate your constructive comments, we have revised our introduction accordingly, it now reads: “It is predicted that the number of hypertensive adults will reach 1.5 billion, which is ~ 30% of the world population by 2025, based on analysis of worldwide data of hypertension burden [1]. Hypertension is notoriously difficult to control, causing detrimental irreversible damage to the important system in the body, e.g. central nerves system and cardiovascular system. In addition to physical impact, psychological disturbance is another major concern among these uncontrolled hypertension patients [2]. Rapid economic development and fast globalization perhaps contribute to the increased stress to the general population via transforming our social organization [3]. The literature has accumulated evaluating the effect of psychological stress in cardiovascular field, including depression and anxiety [4, 5], which are mutually causal and affect each other, and further aggravate the psychological impairment. There are some reports, showing the efficacy of psychological intervention for the management of the patients with comorbidity of hypertension and depression/anxiety. We will explore/validate such findings, using meta-analysis in current study.

It has been reported that substantially increased volume of psychological intervention for the management of hypertension patients with depression and/or anxiety over the last two decades [6]. Depression and anxiety are significant contributors to hypertension at the global level with huge economic burden. The World Mental Health Survey from 17 countries demonstrates that on average about 1 in 20 people reported having an episode of depression in the previous experience [7], suggesting common problem nowadays we face. It has reported that the incidence of depression in hypertension patients reached 20%, which is supported by the finding that cardiovascular disease patients have a higher incidence of psychological disorders, accompanied with increased the risk of cerebrovascular accident in the general population with anxiety and depression disorder [8]. This is supported by the finding from Netherlands for a 5-year study involving the 455,238 women, confirming a close correlation between hypertension and depression/anxiety. Furthermore, depression or anxiety increases 3.5 or 2.0 fold of the risk of hypertension from non-hypertensive people [9]. The finding from Netherlands is consistent with other regions, showing that depression or anxiety increases the risk of hypertension by 1.42 [10] or 1.55 [11] times. Patients with depression and anxiety usually presented with dizziness, headache and chest tightness [12], and often accompanied with negative emotions which impacted quality of life. Such phenomenon is attracted great attention from psychologists recognizing the needs from mental unstable patients [13].” (Introduction, line 57 paragraph 1 page 4)

2.) Please improve citation. Some of the cited references given are insufficient, misleading or incorrect. Examples:

- (1) Hu et al. (2015) is not the original source of the statement given. It'd rather be: Kearney PM, Whelton M, Reynolds K, Muntner P, Whelton PK, He J. Global burden of hypertension: analysis of worldwide data. Lancet. 2005 Jan 15-21;365(9455):217-23. doi: 10.1016/S0140-6736(05)17741-1. PMID: 15652604.

We have citated the suitable references accordingly:

1. Kearney PM, Whelton M, Reynolds K, et al. Global burden of hypertension: analysis of worldwide data. Lancet. 2005,365(9455):217-223. (Reference, line 265 page 15) 

- (6) again the cited source seems to be not the original reference, it'd rather be World Health Organization

An original reference has been cited accordingly:

7. World Health Organization, WHO marks 20th Anniversary of World Mental Health Day. http://www.who.int/mediacentre/news/notes/2012/mental_health_day_20121009/en/. Accessed December 11, 2014. (Reference, line 277 page 15)

- The selection of therapeutic methods (DBT, CBT, MBSR) studied is not sufficiently supported with references. Note that source (17), to which your argument refers, that these treatments are "the most popular psychotherapies depression and/or anxiety in hypertension patients", seems to be a mere study protocol which doesn't mention DBT and MBSR at all.

We apologies for this mistake, and have provided a suitable references supporting DBT and MBSR, it now reads:

“It is reported that mindfulness-based stress reduction (MBSR) [17, 18] and cognitive behavior therapy (CBT) [19] have a good effect in treating chronic diseases patients, and dialectical behaviour therapy (DBT) [20] has a significant effect in management of depression and anxiety.” (Introduction, line 102 paragraph 1 page 6)

20. Fitzpatrick S, Bailey K, Rizvi SL. Changes in emotions over the course of dialectical behavior therapy and the moderating role of depression, anxiety, and posttraumatic stress sisorder. Behav Ther. 2020, 51(6):946-957.” 

- Please note that citation (27), which is supposed to support your statement: "At present, MBSR, CBT and DBT are common psychotherapies in the management of the hypertension patients with depression and anxiety..." refers to Combined Exercise Training, an apparently entirely different form of therapy.

To reduce such confusion, we have selected the reference 26 only as individual therapy, and deleted reference 27 to avoid the combined therapy.

- I could not find a reference to the DBT in your text.

Apologies for this mistake. A suitable reference has been added, it now reads: “It is reported that mindfulness-based stress reduction (MBSR) [17, 18] and cognitive behavior therapy (CBT) [19] have a good effect in treating chronic diseases patients, and dialectical behaviour therapy (DBT) [20] has a significant effect in management of depression and anxiety.” (Introduction, line 102 paragraph 1 page 6)

20. Fitzpatrick S, Bailey K, Rizvi SL. Changes in emotions over the course of dialectical behavior therapy and the moderating role of depression, anxiety, and post-traumatic stress sisorder. Behav Ther. 2020,51(6):946-957.” (Reference, line 382-384 page 16-17) 

3.) You could also slightly sharpen the methodological description: for example, a more comprehensive and precise listing of the inclusion criteria would be useful (definition of the therapy methods studied e.g. length of interventions; inpatient/outpatient; outcomes collected in the included studies and their measurement, minimum sample size; follow-up-timing etc).

We have revised the Inclusion and Exclusion criteria in the Methods accordingly, it now reads: “

2.2. Inclusion criteria

2.2.1. Types of studies 

We will include only randomized controlled trials (RCTs) with MBSR, CBT, DBT.

2.2.2. Types of participants 

1) Hypertension patients with depression or anxiety; 2) Hypertension patients who volunteer for psychotherapy; 3) inpatient/outpatient.

2.2.3. Types of interventions

The eligible interventions include MBSR, CBT, and DBT, the length of each intervention is 8-10 weeks, and the follow-up time is 1-3 months.

2.2.4. Types of comparators 

The control interventions will be included drug therapy and exercise therapy (Yoga, Tai Chi, etc.).

2.2.5. Types of outcomes

The primary outcome indexes of the protocol are depression and anxiety score, and the secondary outcome indexes mainly include blood pressure and other outcome indexes reported in the literature.

2.3. Exclusion criteria

The exclusion criteria include:

1. Hypertension patients who receive any combination of two or more therapies other than MBSR, CBT and DBT.

2. Any hypertension patients treated with any psychotherapy other than the above.

3. Case reports, reviews, abstracts, experimental studies, mechanism discussions, experience summary and other research types of literature.

4. Repeatedly checked or published literature.

5. Incomplete data or information that does not indicate the end result and cannot be included.” (Methods, line 112 paragraph 3 page 6)

The current study is a protocol for a systematic review and Bayesian network meta-analysis. Thus, it will be depended on the studies selected that the length of interventions; inpatient/outpatient; outcomes and measurements, minimum sample size and follow-up-timing.

4.) Since the draft is a registered study protocol, I suspected that the searching and data analysis was still pending. In point 2.4 you write that the information search was finished in December 2021. Is this correct?

Apologies for this mistake, it actually is December 2022. We have modified our manuscript accordingly, it now reads: “A literature search for RCTs will be performed in five electronic databases from inception to December 2022,…” (Methods, line 138 paragraph 3 page 7)

5.) Language: The submitted manuscript contains a number of grammatical errors and lacks comprehensibility in several places. For example, I do not understand the first sentence in the second paragraph of the introduction.

The manuscript has been proofread by a native English speaker. And we have clarified this sentence in the Introduction section, it now reads: “It has been reported that substantially increased volume of psychological intervention for the management of hypertension patients with depression and/or anxiety over the last two decades [6].” (Introduction, line 68 paragraph 2 page 4)

6. Bussotti M, Sommaruga M. Anxiety and depression in patients with pulmonary hypertension: impact and management challenges. Vasc Health Risk Manag. 2018,14:349-360.

Reviewer #2:

First, you should use proper English.

The manuscript has been proofread by a native English speaker.

Using "will" is inappropriate in most parts. It is appropriate to use the past tense for what you have analyzed.

The current study is protocol, the research has not started when we designed and wrote it.

You should also avoid writing in a way that makes it unclear whether it is your idea or a cited finding. In particular, there are a number of sentences that are described as general findings but do not have citations attached.

We have modified accordingly for the full manuscript.

Major point

1. Introduction in abstract

The following is an exaggeration, because it is stated as if all hypertensive patients are suffering from depression and anxiety as well. “However the outcomes of medications maybe compromised in some individuals without following medical instruction, partially due to these hypertension patients are co-exist with depression and/or anxiety.”

We have modified this point in the Introduction section, it now reads: “Due to quite large number of hypertension patients co-exist with depression and/or anxiety, and non-cooperated with medical instruction, consequently management of BP is impaired with serious complications, resulting in compromised quality of life.” (Abstract, line 23 paragraph 1 page 2) 

The following leads to the erroneous conclusion that measures to address the risk factor will improve hypertension. “Because depression/or anxiety is an independent risk factor for hypertension, stress relieving would be an ideal management of hypertension. Psychotherapy for hypertension patients with depression and/or anxiety improves psychological symptoms and controls BP.”

We have modified this point, it now reads: “Depression and/or anxiety are independent risk factors of hypertension, which is supported by the finding that there is close correlation between hypertension are depression/or anxiety. Psychotherapy (non-drug treatment) maybe useful for hypertensive patients with depression and/or anxiety to improve their negative emotions.” (Abstract, line 28 paragraph 1 page 2)

2. Introduction

Please add a reference to the following sentence.

“In addition to physical disability, psychological disturbance is another major concerns among these uncontrolled hypertension patients.”

We have added the reference accordingly.

2. Ovbiagele B, Nguyen-Huynh MN. Stroke epidemiology: advancing our understanding of disease mechanism and therapy. Neurotherapeutics. 2011,8(3):319-329. (Reference, line ** paragraph * page *) 

“Rapid development and fast globalization perhaps contribute to the increased stress level via transforming our social organization.” From the findings of the literature in [3], the following statements are clearly overstated. “It has been demonstrated that significantly increased psychological intervention for the hypertension patients with depression and/or anxiety over the last two decades [3].”

We have modified our statement, it now reads: “Rapid development and fast globalization perhaps contribute to the increased stress level via transforming our social organization [3]. The literature has accumulated evaluating the effect of psychological stress in cardiovascular field, including depression stress and anxiety stress [4, 5].” (Introduction, line 61 paragraph 1 page 4) 

3. Yu B, Chen X, Li S. Globalization, cross-culture stress and health. Chin J Epidemiol. 2014,35(3):338-341.

4. Zhang Y, Chen Y, Ma L. Depression and cardiovascular disease in elderly: Current understanding. J Clin Neurosci. 2018, 47:1-5.

5. Cohen BE, Edmondson D, Kronish IM. State of the Art Review: Depression, Stress, Anxiety, and Cardiovascular Disease. Am J Hypertens. 2015 ,28(11):1295-1302.

3. Method

For example, please specify the Study protocol in the following form:

[The protocol of the present study was registered in the international prospective register of systematic reviews]

We have modified it accordingly, it now reads: “

2.5 Register

The protocol of the present study was registered in the international prospective register of systematic reviews (Prospero registration number: CRD42021248566).” (Methods, line 144 paragraph 4 page 7)

You have narrowed your criteria for depression and anxiety to two: the PHQ-9 depression scale and the GAD-7 anxiety scale. Do you have any clear explanatory evidence that this does not constitute selection bias?

We have modified it accordingly, it now reads: “We will utilize the data available from the selected literatures, but not only narrow our study into the PHQ-9 depression scale and the GAD-7 anxiety scale. The primary outcomes are the modified depression and/or anxiety scores, by comparing prior to and post intervention. We will further compare the blood pressure prior to and post intervention, as objective evidence for improving the patients’ conditions.” (Methods, line 176 paragraph 2 page 10) 

4. Results.

?There is no results chapter.

For example, please describe in the following way.

How many cases were included in the inclusion criteria, how many were excluded, and how the results differed for each study.

Apologies for this mistake. There is no result in the current protocol yet. The case number will be quantified in our future study.

5. Discussion

Discussion is meaningless because there is no result.

Again, we apologies for this mistake. This is no full discussion in the protocol. Instead of discussion, we summaries our protocol in this section with limited highlights and limitation.

6. Figure

Please create a figure of the literature search flow chart showing the number of literature hits for each literature search method. If you create a figure, please also create network diagrams of comparisons.

We apologies for this mistake, however the study is only a protocol, and the number of literature hits for each literature search method will be shown in the next step of study.

Minor point

AGREE II (The Appraisal of Guidelines for Research & Evaluation II) and other tools, and consider whether a systematic review is sufficient in the first place or whether a meta-analysis is necessary.

We believe that a meta-analysis is needed for the current study after careful evaluation.

7. PLOS authors have the option to publish the peer review history of their article (what does this mean?). If published, this will include your full peer review and any attached files.

We choose yes for this point

Do you want your identity to be public for this peer review? For information about this choice, including consent withdrawal, please see our Privacy Policy.

Reviewer #1: No

Reviewer #2: No

Acknowledged.

We have revised our manuscript accordingly, and hope our manuscript meet the standard for publication in PLOS ONE.

Yours sincerely

Jingchun Fan

---

## [Editor Report · Decision Letter 1]

25 Jan 2023

Effect of MBSR, DBT and CBT on the hypertension patients with depression/anxiety: Protocol of a systematic review and Bayesian network meta-analysis

PONE-D-21-36202R1

Dear Dr. Zhang,

We’re pleased to inform you that your manuscript has been judged scientifically suitable for publication and will be formally accepted for publication once it meets all outstanding technical requirements.

Kind regards,

Stephan Doering, M.D.

Academic Editor

PLOS ONE

---

## [Editor Report · Acceptance letter]

10 Feb 2023

PONE-D-21-36202R1 

Effect of MBSR, DBT and CBT on the hypertension patients with depression/anxiety: Protocol of a systematic review and Bayesian network meta-analysis 

Dear Dr. Zhang:

I'm pleased to inform you that your manuscript has been deemed suitable for publication in PLOS ONE. Congratulations! Your manuscript is now with our production department. 

Kind regards, 

on behalf of

Professor Stephan Doering 

Academic Editor

PLOS ONE